# Structure and Frictional Properties of the Leg Joint of the Beetle *Pachnoda marginata* (Scarabaeidae, Cetoniinae) as an Inspiration for Technical Joints

**DOI:** 10.3390/biomimetics5020014

**Published:** 2020-04-20

**Authors:** Steffen Vagts, Josef Schlattmann, Alexander Kovalev, Stanislav N. Gorb

**Affiliations:** 1Department of System Technologies and Engineering Design Methodology, Hamburg University of Technology, Denickestr. 22, D-21079 Hamburg, Germany; j.schlattmann@tuhh.de; 2Department of Functional Morphology and Biomechanics, Kiel University, Am Botanischen Garten 9, D-24118 Kiel, Germany; akovalev@zoologie.uni-kiel.de (A.K.); sgorb@zoologie.uni-kiel.de (S.N.G.)

**Keywords:** locomotion, walking, leg, joints, insects, Arthropoda, friction, coefficient of friction, biotribology, biomimetics

## Abstract

The efficient locomotion of insects is not only inspiring for control algorithms but also promises innovations for the reduction of friction in joints. After previous analysis of the leg kinematics and the topological characterization of the contacting joint surfaces in the beetle *Pachnoda marginata,* in the present paper, we report on the measurement of the coefficient of friction within the leg joints exhibiting an anisotropic frictional behavior in different sliding directions. In addition, the simulation of the mechanical behavior of a single microstructural element helped us to understand the interactions between the contact parts of this tribological system. These findings were partly transferred to a technical contact pair which is typical for such an application as joint connectors in the automotive field. This innovation helped to reduce the coefficient of friction under dry sliding conditions up to 17%.

## 1. Introduction

In tribology, the coefficient of friction μ plays the most important role in characterizing the systems properties. Amonton (1699) and Coulomb (1785) summarized their findings on the friction between two solid bodies in the well-known Equation (1). This relationship describes the proportionality of the tangential force F_t_ to the normal force F_n_ by the proportionality factor μ.
(1)$$\mu =\frac{F_{t}}{F_{n}}$$

In principle, a distinction is made between the static friction and the dynamic friction or the static μ_s_ and the dynamic μ_k_ coefficients of friction. The static frictional force F_s_ must be overcome in order to set two contacting objects in motion. The force, which acts opposite to the direction of movement, is referred to as the sliding friction force F_f_. The resistance to movement is usually caused by interactions between the molecules of the contacting surfaces, but the mechanical interlocking between asperities of both contacting surfaces can also contribute [1,2].

However, this equation implies an independence of the sliding frictional force F_f_ from the real contact surface and the surface roughness, which has been the subject of considerable controversy for a long time. This claim can be confirmed by studies that considered macroscopic frictional contacts [1,3,4,5]. However, this fails for the consideration of friction at the micro- and nanoscale, which has also been confirmed by many studies [1,6,7]. It is assumed that the frictional force on a nanoscale also depends on adhesion [1]. The downsizing of tribosystems leads to an increase in the role of factors influencing friction, adhesion, and wear between the contacting surfaces [8]. When the normal force is in the range of mN and μN, capillary, electrostatic, and Van der Waals forces play not a negligible role but instead a dominant role in their contribution to friction [2,9,10].

The measurement of friction coefficients has already been carried out for many biological systems [11], but most tribologically specialized biological systems are joints. On the other hand, while vertebrate joints are rather well studied, tribological properties of insect joints remain largely unknown. Dai and Gorb [12] presented the first study about microstructural adaptations and frictional characteristics in beetles’ joints. Kheireddin et al. [13] performed a tribological study on the leg joint of a grasshopper (*Romalea guttata*) and revealed a very low coefficient of dynamic friction μ_k_ between the condyles of the femorotibial joint. Dai and Gorb [12] and Barbakadze [14] examined the head thorax joint of the Congo rose beetle (*Pachnoda marginata*) on its sliding friction. For this purpose, the friction was measured between the gula plate (joint surface of the head) and a glass plate during the relative movement between the contact partners. In these investigations, no directional dependence of the coefficient of friction could be determined. On the other hand, anisotropic friction has been previously demonstrated in studies on other biotribological systems, such as the abdominal scales of snakes [15,16,17]. The microstructure elements there have a strongly asymmetrical cross-sectional shape and are similar to those present on the tibial joint surface of *P. marginata* [18]. Vagts et al. [18,19,20] investigated the surface topology and the kinematic relations in the leg joints of *P. marginata*. They have approved the following hypotheses. (1) The contact pair consists of convex and concave geometries, which geometrically fit each other. (2) There is always a smooth and a rough surface in contact with both corresponding counterparts of the joint, especially the tibia–tarsus joint of *P. marginata*, which shows a surface topology that is characterized by microstructural elements [18]. In addition, indentation tests showed that contacting parts in leg joints of the beetle show a significant difference in elastic modulus and hardness [18].

With regard to this previous research work, the following new hypotheses were formulated in the present paper:The coefficient of sliding friction µ_k_ has a dependence on the real contact area for a low contact stress;There is a directional dependence of the μ_k_ that is given by the microstructure with an asymmetric cross-section.

Additionally, for a better understanding of the behavior of the tribological system under consideration, the material architecture (i.e., layering, fiber orientation, material density) of both partners of the contact pair was further analyzed [19]. To understand the structural origin of the mechanical properties of insect cuticles, which in general is a hierarchical material composite, the layered cuticular structures of the corresponding parts under consideration were investigated in detail. This analysis served as the foundation for a further finite element analysis (FEA) for the contact mechanical simulation of the microstructure-covered tibial–tarsal joint of *P. marginata*. Due to the FEA, we obtained information about the deformation behavior of an individual microstructure element, in order to interpret the results of the sliding friction measurements carried out in the context of the previously formulated hypotheses. These findings were further partly transferred to a technical contact pair that is typical for a joint connector in the independent wheel suspension in a car to fulfill a friction reduction in sliding contact. The samples were machined by a selective laser melting technology and compared with non-structured samples of the same material pair.

## 2. Materials and Methods

After analyzing the kinematic properties of legs [20] and the identification and topological characterization of the contacting joint surfaces and their microstructural adaptations in the beetle *Pachnoda marginata* [18], the next step was to look into the tribological properties of this biological contact system. The present paper comprises the measurement of the coefficient of friction within the leg joints of *P. marginata*. In addition, the simulation of the mechanical behavior of a single microstructural element helped us to understand the interactions between the contact parts of the tribological system [19].

### 2.1. Sample Preparation of Insects

The beetles, *Pachnoda marginata* (*n* = 5), were taken from the laboratory colony at the Kiel University. For the preparation, the relevant leg segments were first separated with scissors. The respective joint was opened with a microscalpel and the two leg segments were separated. Subsequently, the surfaces of the joint were exposed, separated, and fixed on a glass slide holder using an adhesive (5925 Elastomer, Ergo, Kissing, Switzerland). For the friction measurements, the beetles’ leg joint segments were prepared fresh right before the experiment.

### 2.2. Scanning Electron Microscopy of Insects

To examine the ultrastructure of the cuticle of the tibial–tarsal joint, dried joint parts (24 h in room air) were prepared. Due to the high grade of sclerotization of the joint cuticle, a mechanical fracture with a microscalpel provided a cuticle cross-section suitable for the microscopy analysis. Scanning electron microscopy (SEM) was used to visualize the fractures of the contacting joint surfaces. The prepared samples were fixed on an aluminum sample holder and sputtered with gold–palladium (layer thickness: 10 nm). For the visualization of the fractured cross-sections, a Hitachi S-4800 SEM was used, operated at an acceleration voltage of 3 kV.

### 2.3. Confocal Laser Scanning Microscopy

In the present study, the surfaces of the beetles’ joints were analyzed using a Keyence VK-8710 confocal laser scanning microscope (Keyence Corporation, Osaka, Japan). Two brightfield objectives (Nikon CF IC EPI Plan Achromat 20×, numerical aperture = 0.46, working distance = 3.1 mm; Nikon CF IC EPI Plan Achromat 50×, numerical aperture = 0.8, working distance = 0.54 mm) were applied. The preparation was exposed to light from a stable solid-state laser with a wavelength of 635 nm (5 mW at the fiber end, laser power = 2%), and the laser light reflected at the surface of the joint surfaces was detected. The detector gain was automatically adjusted prior to the image stack collection in a way that resulted in a maximum signal intensity with the simultaneous prevention of oversaturation. The image size was set to 1024 × 1024 pixels and the scan step size was 0.5 μm for 20× magnification and 0.1 μm for 50× magnification. The software VK-analyzer was used to create the maximum-intensity projections and color-coded height maps.

### 2.4. Friction Measurements of Beetle Joints Using Nanotribometer

For the tribological exploration of the beetles’ surfaces in the nanometer and micrometer range, we used an NTR^2^ nanotribometer (Anton Paar, Switzerland) with a maximum force resolution of 3 nN. It was mounted on an air-cushioned table with a granite plate to protect against external vibrations. The device was equipped with a high-resolution cantilever which has a spring stiffness of c_n_ = 602 N/m in the normal direction and c_t_ = 170.5 N/m in the tangential direction.

With a piezo-actuated table, the counter sample can be moved by a maximum of Δl = 0.5 mm relative to the base sample. The normal force is applied through the base sample, which can be lowered by a piezoactuator by a maximum of 0.5 mm. In order to study the microstructures in terms of their importance for friction maintenance, the tibial–tarsal joint of *Pachnoda marginata* was examined with respect to its coefficient of sliding friction μ_k_. For this purpose, the tangential force F_t_ was measured between the condyles of the tibia and tarsus while these are moved with a constant sliding speed of v = 0.025 cm/s and loaded with normal forces of 0.5, 1, 2, 5 and 10 mN. These forces are applied three times in succession for 40 cycles each, so that 15 measurements per individual tibial–tarsal joint were performed. The sliding amplitude of each tarsus sample was about 80 μm, which oscillated around a fixed central position that was in the center of the tibial condyl (total sliding distance: 3.2 mm). The ambient conditions during the measurements were a room temperature of 21.5−23.5 °C and a relative humidity of 20%–30%.

### 2.5. Friction Measurements of Bioinspired Joints Using Tribometer

The friction coefficient measurements on the bioinspired joints were carried out with a universal test bench from Zwick AG in the house of Daimler AG in Harburg, Hamburg, with a pin-on-disk (drilling) model system. The base body (sphere of the ball joint) consisted of the tempered steel 42CrMo4 (material number 1.7225). The counter body (plate) was made from the material Victrex PEEK 150CA30. There was no intermediate medium for the measurements (dry sliding conditions), and an angular velocity of 10°/s was selected. Before and after the friction measurements, samples were cleaned in an ultrasonic bath for five minutes at 50 °C and degreased with Rivolta (A.C.S. Spray). The normal forces were increased stepwise from 0.5, 1.0, 1.5 to 5.0 kN.

### 2.6. Finite Element Analysis

The mechanical properties of the beetles’ joint cuticle were modeled using the finite element method (FEM). For the modeling of the exocuticle, a transversal isotropic material behavior was assumed. For the simulation of cuticle deformation behavior, a transverse isotropy was used, since fiber composites normally have at least one orthotropic plane [21] and also since the Bouligand Helix model uses a transverse isotropy. After characterizing the mechanical material behavior of the cuticle, the deformation behavior of a single microstructure element was simulated by means of a static, non-linear structure FEM with contacts using Abaqus software (Version 6.13-1, Dassault Systems, Velizy-Villacoublay, France). Based on the transversal isotropic material behavior according to Equation (2), the input parameters for the engineering constants material model shown in Table 1 were used. The purpose of the FEA is a qualitative investigation of the mechanical deformation behavior of a single microstructural element under normal load and not the simulation of the friction coefficient between the tibia and tarsus in different directions. The values of elastic modulus E3 were previously measured using nanoindentation [18] and used for the simulation of the realistic asymmetric contact behavior of the tibia and tarsus of *P. marginata*.

### 2.7. Preparation of Bioinspired Surfaces Using Selective Laser Melting

The prototype consists of tempered steel 42CrMo4 (material number 1.7225) with a diameter of 10 mm. The contact surfaces of all samples were randomly grounded with a target roughness of Rz < 3 µm and were quality controlled. The microstructural machining was carried out using a laser line system from Ewag AG, which was equipped with the picosecond laser Hyper Rapid 50 (power: 50 W, pulse length: 12 ps) from Lumera Laser GmbH. The microstructures were laser micromachined to a planar surface of 42CrMo4. The realized microstructures were not asymmetric because the real-world application of that tribosystem in the ball joints of the independent wheel suspension of a car does not require anisotropic friction behavior, but rather an improved coefficient of friction and failsafe running functions.

### 2.8. Statistical Analysis

The statistical analysis of the measurement data was performed with IBM SPSS Statistics (Version 22, Armonk, NY, USA). The non-parametrical analysis of variance was computed using the Kruskal–Wallis test by ranks.

## 3. Results

### 3.1. Results of Biological Contact Pair

Figure 1D–G shows the cross-section of the fractured tibial–tarsal joint surfaces of the beetle. Table 2 summarizes the thickness of the individual cuticle layers. The endocuticle is characterized by a microstructure in which the fiber bundles are layered in a 0°/90° orientation. The fiber bundles running perpendicular and parallel to the fracture plane are clearly visible in Figure 1D–G. It also shows that the articular surfaces, whose cuticle ends in the medial direction in the interior of the leg segment and thus adjoins the internal space filled with hemolymph, form no further exocuticle layer (Table 2). The thickness of the exocuticle ranges between 12.0 and 22.5 μm. The thickness of the endocuticle at many sites is several times thicker than the exocuticle and ranges between 56.0 and 73.0 μm.

The layered structure of the exo- and endocuticle was revealed for the contacting surfaces of the joint. This hierarchical material architecture with different layers and fiber orientations is important to consider in the analysis of the contact–mechanical interactions of both contact partners. Both the exocuticle, with its thin multiple layers, and the thin and chemically inert epicuticle are present at the joint interface, in order to protect the material against external influences.

Figure 2 show the measured forces in the tangential F_t_ and normal F_n_ directions for the fresh and dried samples. It can be clearly seen that the shear of the dry samples causes lower tangential forces than in the fresh contact pair. The difference increases with the increase in normal forces. The difference of the tangential force F_t_ with the increasing normal force F_n_ is more pronounced in the fresh sample than in the dried sample. To compare the difference between the tangential forces F_t_ of the fresh and dry samples as a function of the normal force levels F_n_ = 0.5, 1, 2, 5 and 10 mN, the Mann–Whitney rank sum test was performed. For the proximal sliding direction, at F_n_ = 2 mN the tangential forces of the two sample conditions differ significantly from each other (σ < 0.05). For the distal direction, a significant difference in the tangential forces for the fresh and the dry contact pair only occurs after a normal force of F_n_ = 5 mN (σ < 0.05). Figure 3 shows the plot of the coefficient of sliding friction μ_k_ as a function of F_n_ of the fresh and the dried samples of the tibio–tarsal joint. Here, it becomes clear that the μ_k_ depends on the normal force. For the dry contact pair, the coefficient of sliding friction decreased from approximately 0.15 at a normal force of 0.5 mN to approximately 0.10 at 5 mN and then remained rather constant. The resulting coefficient of friction in the proximal direction (µ_k,p_ = 0.098 ± 0.021) is not significantly smaller than in the distal direction (µ_k,d_ = 0.110 ± 0.031, Mann–Whitney rank sum test (σ > 0.05)).

In the fresh sample condition, this relationship was less pronounced. Although the μ_k_ for the proximal direction dropped from approximately µ_k_ = 0.175 at F_n_ = 0.5 mN to 0.15 at a normal force of 1 mN, the value remained constant for the increasing normal forces. For the distal direction of the fresh sample, there was no drop in the μ_k_ over the increasing normal force, but rather an increase of approximately 0.175 at 0.5–2 mN to approximately 0.185 at 5–10 mN normal load. The resulting coefficient of friction in the proximal direction (µ_k,p_ = 0.153 ± 0.015) is significantly smaller than in the distal direction (µ_k,d_ = 0.200 ± 0.014, Mann–Whitney rank sum test (σ < 0.05)).

The coefficient of sliding friction μ_k_ of the fresh samples, both in the proximal and distal directions, was about 36% greater than that of the dried samples. The slopes of the proximal and distal sliding directions differ by approximately 12% in both sample conditions.

Figure 4 represents the simulated deformation of the microstructure element (B) located in the articular surface of the tibia compared to the initial situation (A) by nodal displacement in the normal surface direction. In this case, the steep (proximal) flank of the element is compressed and the flat flank (distal) approaches the cuticle of the tarsus. Figure 5 shows in detail the stepwise deformation behavior of a microstructure element. It becomes clear that the angles α and β are changed by the deformation (α_start_ > α_end_; β_start_< β_end_, see Figure 4). Due to the significant difference in the elastic modulus of the exocuticle of the tibia and tarsus, the counter body (tarsus part) is deformed less than the base body (tibia part with its microstructure). The deformation behavior and the dentoid shape of the microstructure imply that sliding in different directions occurs on the one hand along the flat flank (in proximal direction) and on the other hand along the steep flank (in distal direction).

### 3.2. Results of Technical Contact Pair

Figure 6 shows the machined surface of the prototype pin (B) in comparison to a non-structured sample (A) for the dry friction measurement. The realized microstructuring was designed for a reduction in friction in the application as a joint connector in the independent wheel suspension of a car. We did not transfer anisotropic friction properties of the biological system to the technical samples because the ball joints do not require anisotropic friction behavior. The percentage of the contact surface was reduced to 31% in comparison with a non-structured contact surface. However, strongly reduced friction (up to 17%) and improved failsafe running functions are reached by the spatial arrangement of the microstructural elements inspired by *P. marginata*. Special attention was paid to the avoidance of melt build-up or burr formation on the edges of the microstructure element as a result of the ablation process, since this would lead to a strong roughening of the contact surface. The evaluation of the realized geometric dimensions of the microstructure was carried out by confocal laser scanning microscopy and is shown in Table 3 and Figure 6C. Figure 7 summarizes the results of the force measurements of the textured pin-on-disc and the non-textured pin-on-disc tribopair. Here, the coefficient of friction of the structured sample (µ_k_ = 0.317 ± 0.011) is significantly smaller than of the non-structured sample (µ_k_ = 0.360 ± 0.015, Mann–Whitney rank sum test (σ < 0.05)). The friction coefficient between the PEEK and a steel disc (surface roughness Ra = 1.34 μm) is 0.32 at a PV (pressure x velocity) value (Pa m/s) of 149.690 [24]. On the basis of the comparable order of magnitude of the calculated coefficients of friction, the results can be regarded as reliable despite the significantly higher PV values (425,000 to 4,245,000). Due to the microstructuring, the coefficient of friction could be reduced by between about 9% and 17% compared with the unstructured surfaces.

## 4. Discussion

### 4.1. Biological Tribo System

The cuticle is generally subdivided into three main layers, namely the epi-, exo-, and endocuticle, each main layer consisting of further sublayers [25,26,27] (Figure 1). The exo- and endocuticle together form the procuticle. The epicuticle is the outermost layer of the cuticle and is very thin (1–2 μm) [25]. It consists of cement and wax sublayers, as well as inner and outer epicuticular layers. Each sublayer has a different composition and properties and does not contain chitin. These structures often create diffraction patterns and functional friction surfaces with special properties. The procuticle is the chitin-reinforced part of the cuticle. In addition to proteins and chitin, such important additional components as water, polyphenols, and lipids are present there. The thickness of this layer varies depending on the insect species and the body part of the insect in the range of 1 µm to more than 200 µm and makes up the major part of the cuticle [26]. Chitin is a polysaccharide that occurs in insects in the form microfibril (25–30 Å in diameter, 300 nm in length) bundles. The individual fibrils are then present in large fiber bundles and form a typical 0°/90 ° interchanging orientation in the endocuticle. The matrix consists of a rubbery protein (resilin), which has variable mechanical properties depending on the water content. The structure of the procuticle, with its complex pattern of microfibril orientations, is very similar to that of a technical fiber composite material [26,27]. However, the fibers within the planes are not randomly oriented. Among other explanatory models for the ultrastructural architecture of the procuticle, there is the Bouligand helix model, which describes the following structure:The microfibrils are situated within planes parallelly orientated to the cuticle surface;Within one layer, all chitin fibers run parallel to the surface and to each other;Microfibrils situated in successive levels are usually rotated relative to each other at an angle that can vary or remain constant in different cuticles;The number of layers creating full rotation (180°) are called lamella. The thickness of lamella varies depending on the thickness of the microfibrils and the angle of rotation.

In this special case of orthotropic material composition, there is a preferred direction in which the material behaves differently under load than in the other spatial directions that are perpendicular to the preferred direction (Figure 8). The two spatial directions, which are perpendicular to the preferred direction, span an isotropic plane in which the material behavior is independent of the direction. This means that changing the direction of loading in the isotropic plane does not change the material behavior [21]. With the assumption that the layers are thin in relation to their length and width, have a constant thickness and a constant cross-section, a shift of the layers does not occur with each other, and the deformations occurring are small and linear elastic, it is possible to reduce the independent elastic characteristics for formulation of the material behavior on five characteristic values [28]. Zhao et al. [29] investigated the alignment of molecules in polymer networks and showed the influence of this alignment to mechanical properties like elastic modulus. By considering the layers as homogeneous and by the orthotropic material behavior, the following identities of the moduli of elasticity (E_1_, E_2_, E_3_), shear moduli (G_12_, G_23_, G_13_), and Poisson ratios (n_12_, n_23_, n_13_) in the 1,2,3-coordinate system result in Equation (2) [30]:(2)$$E_{1}=E_{2},G_{23}=G_{31},\frac{\nu_{12}}{E_{1}}=\frac{\nu_{21}}{E_{2}},\frac{\nu_{13}}{E_{1}}=\frac{\nu_{23}}{E_{2}},G_{12}=\frac{E_{1}}{2\left(1+\nu_{21}\right)}$$

For the simulation of the cuticle’s deformation behavior, a transverse isotropy is used, since fiber composites normally have at least one orthotropic plane [21] and also since the Bouligand helix model has a transverse isotropy. In addition, the results of the FEA show that the deformation behavior of a microstructure element in the tibial–tarsal joint has a significant influence on the friction between the two articulating surfaces. Due to the superposition of the friction coefficients of different scales [5], the anisotropic friction behavior can be explained by the respective edge slopes α and β of the microstructural elements [26]. However, as the helix is intrinsically chiral, the direction-dependent anisotropic of friction might be affected by helix chirality. The measured anisotropy of the coefficient of sliding friction for the distal and proximal sliding directions accompanies the findings of Baum et al. [17] from the study of snake belly scales of the Californian chain snake (*Lampropeltis getula californiae*). Interestingly, this effect seems to cause about a 12% higher energy dissipation by friction on a nominally flat surface, contrary to the microstructured surface.

The curves of the sliding friction coefficient μ_k_ of the fresh and dry sample are decisively conditioned by the different mechanical properties of the contact surfaces in the course of desiccation. In this paper, we did not study the shrinkage of the cuticle by desiccation directly. However, from other literature sources we know that the shrinkage of highly sclerotized areas of the cuticle is not strong. The shrinkage can cause some additional waviness in the surface (as previously reported by Barbakadze et al. [14] and Barbakadze et al. [31]) due to the stronger deformations of the inner layers (stronger-hydrated layers). However, the superficial layers, which are responsible for the formation of the microstructures, which were studied in the paper under consideration, almost do not shrink. These shrinkages are usually at the nanoscale as it was previously shown by studies on the optical properties of cuticle layers [32]. The mechanical adaptability of the contact partners is a significant factor influencing the adhesion, since it affects the size of the true contact surface [9,33]. The influence of the adhesive contact process at low normal forces of F_n_ = 0.5–2 mN, which has already been demonstrated by Barbakadze et al. [14], clearly shows in the dry sample condition that the real contact surface is sustainably reduced due to the increased material stiffness/hardness in combination with the pronounced microstructuring. Adhesion of contacting solids is caused by attractive atomic interaction forces [9]. As a result, the adhesive contact of a microstructured surface can be significantly reduced [34,35]. It has already been shown that the adhesion is influenced not only by the height of the surface asperities but also by the thickness of the absorbed liquid layer at the contact point [36], which causes the higher coefficient of sliding friction of the fresh sample for low normal loads. Due to the normal forces in the range of mN, capillary forces from the surface water on the contact and electrostatic attraction also contribute to friction. In this paper, we showed that biomimetic microstructuring inspired by beetle joints can potentially reduce friction in the technical joints. With an increasing normal force, the real contact surface and the influence of the real contact area on the friction also increases. This relationship is also attenuated in the proximal direction (along the microstructure) of the fresh contact pair. The absence of this effect in the distal direction might be due to the interlocking of the microstructures, whose reduced stiffness dominates at the contact formation even at the lowest normal forces [37].

### 4.2. Artificial Tribo System

Another explanation for the reduction in the tangential force could also be the trapping of wear particles in grooves that lead to avoiding additional surface plowing. These grooves prevent the accumulation and connection (agglomeration) of particles (e.g., dirt and wear) in the interface between the two friction partners, which in extreme cases could even lead to a blockade of the technical joint parts [38,39,40]. It can be concluded that the optimal dimension of the friction-reducing microstructures should be selected in a way that the reduction of the real contact area of the tribo-pair is fulfilled without causing mechanical interlocking [7]. In addition, it has already been shown that the dimension of the microstructures has an influence on the avoidance of the stick–slip effect [41] and elimination of static friction.

## 5. Conclusions

Since we show a rather low coefficient of sliding friction in the beetle’s joints, we assume a strong influence of the surface microstructure on the improved tribological properties of the joint. There is also a directional dependence of the μ_k_ given by the asymmetric microstructure. The curves of the sliding friction coefficient μ_k_ of the fresh and dry samples are decisively caused by the different mechanical properties of the contact surfaces. The deformability of the contact partners is a significant factor influencing the true contact area and, together with the asymmetric shape of the microstructural elements, causes the anisotropic frictional properties. This, in turn, can make further contributions to the expansion of the functional properties for technical joints and devices [42,43,44].

## Figures and Tables

**Figure 1 biomimetics-05-00014-f001:**
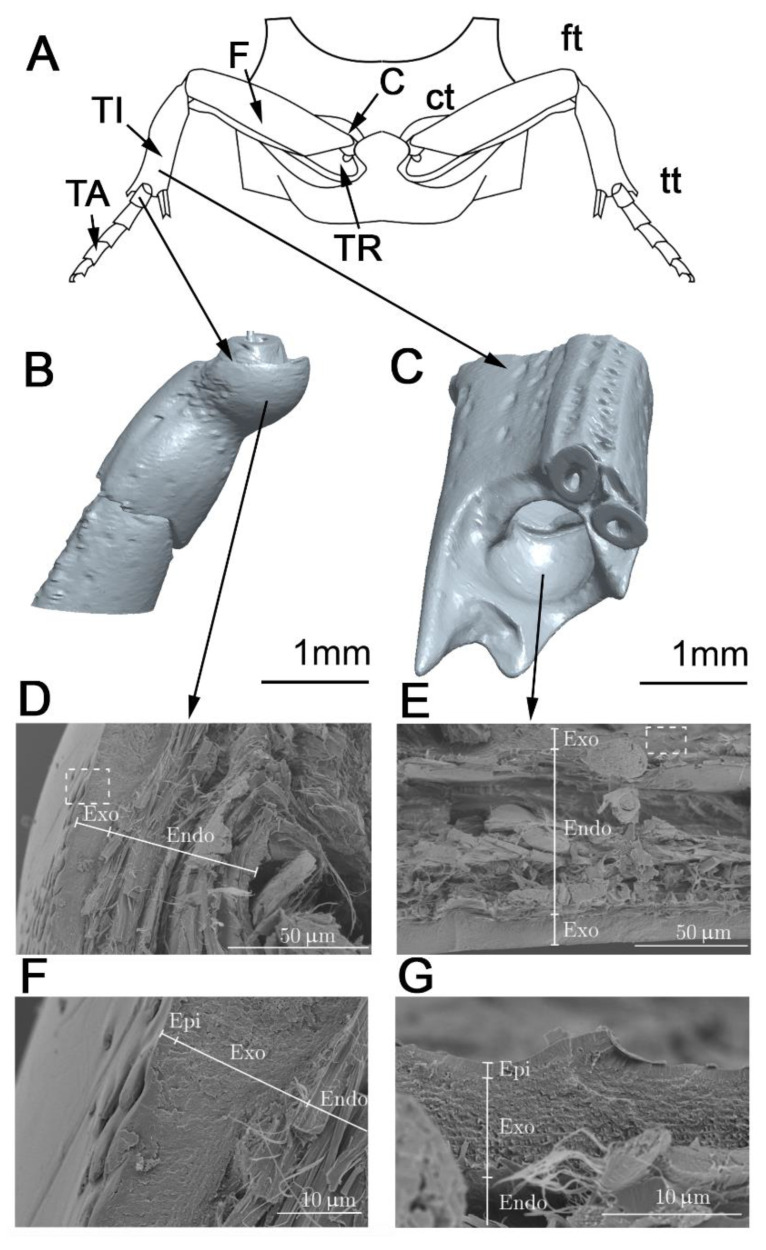
The mesothoracic leg pair of *P. marginata* (**A**) and the tibia (**C**) and tarsus (**B**) segmented from µCT images. Scanning electron microscopy (SEM) images of the cross-fracture of the contacting surfaces in the leg joint between the tibia (**E**,**G**) and tarsus (**D**,**F**) in the beetle *P. marginata*. The dashed box indicates the location of the detailed images F and G for the overview D and E. C: coxa; TR: trochanter; F: femur; TI: tibia; TA: tarsus; ct: coxa-trochanter; ft: femur-tibia; tt: tibia-tarsus joint; Epi: epicuticle. Exo: exocuticle. Endo: endocuticle.

**Figure 2 biomimetics-05-00014-f002:**
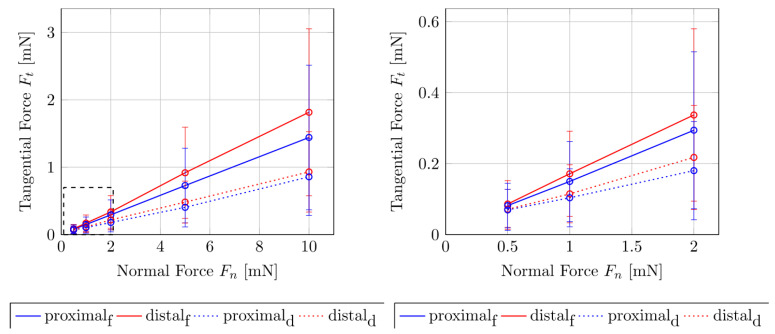
Tangential and normal forces measured in the tibia–tarsus joint of *P. marginata* by nanotribometer NTR^2^. Error bars show standard deviations of the means. The dashed box indicates the detailed plotted data on the right. Distal: in the distal sliding direction. Proximal: in the proximal sliding direction. Index f: fresh sample condition. Index d: dry sample condition.

**Figure 3 biomimetics-05-00014-f003:**
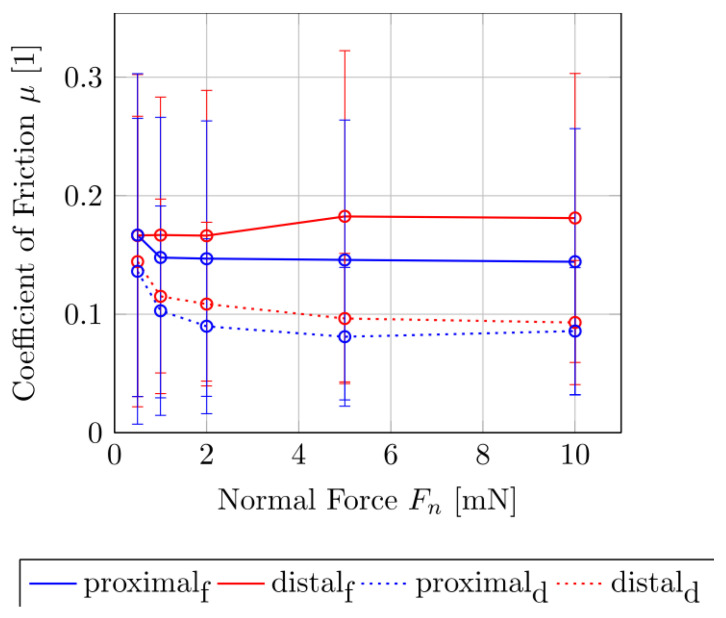
Coefficient of friction for different normal loads calculated from friction force measurements in the tibia–tarsus joint of *P. marginata*. Error bars show standard deviations of the means. Distal: in the distal sliding direction. Proximal: in the proximal sliding direction. Index f: fresh sample condition. Index d: dry sample condition.

**Figure 4 biomimetics-05-00014-f004:**
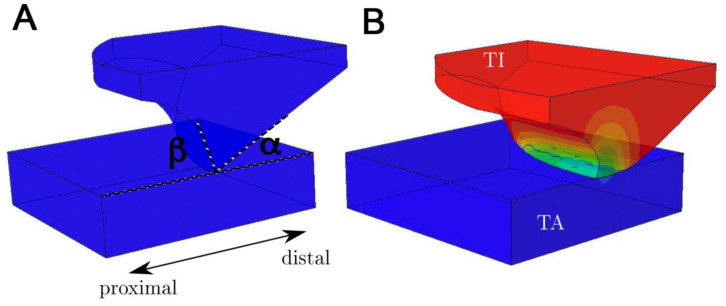
Image of the deformation result of the microstructural element in tibia–tarsus joint of *P. marginata* simulated by a finite element analysis. (**A**) shows the initial shape of the element whereby (**B**) presents the qualitative deformation after a 1.2 µm displacement of the tibia (red) towards the tarsus (blue). TI: tibia; TA: tarsus; α: distal flank angle; β: proximal flank angle.

**Figure 5 biomimetics-05-00014-f005:**
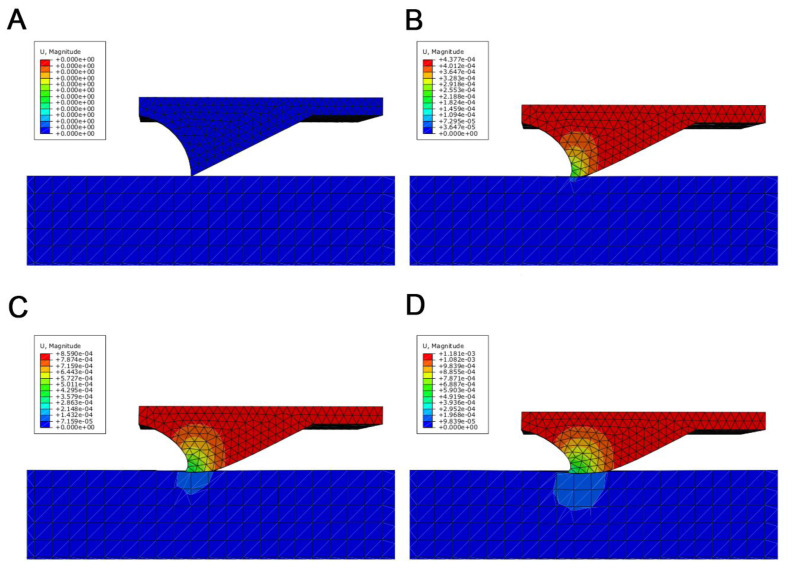
Stepwise nodal displacement (**A**–**D**) of the microstructure element located in the tibial–tarsal joint of *P. marginata*. Finite element analysis was performed using Abaqus finite element method (FEM) software with transversal isotropic material model. Displacement units are given in millimeters. The joint segments and orientation are identical to the labeling in Figure 5.

**Figure 6 biomimetics-05-00014-f006:**
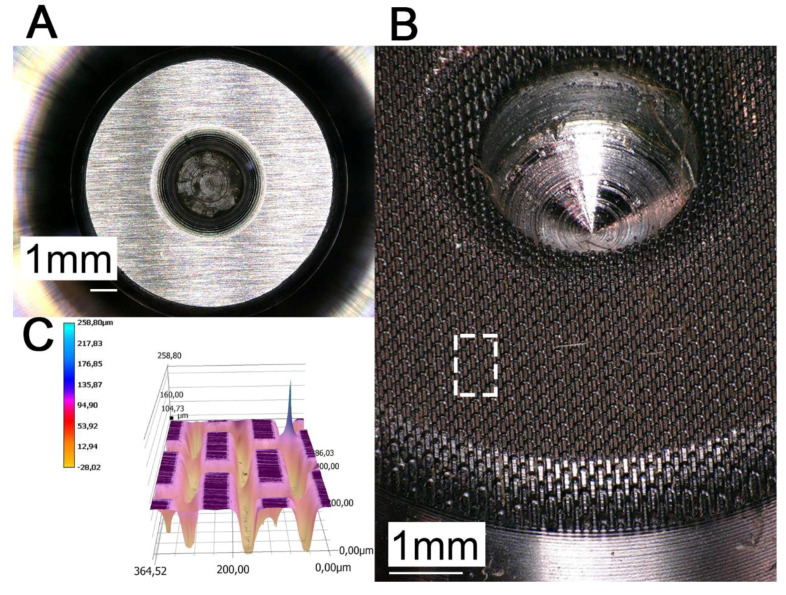
Images of surface topology of non-textured (**A**) and textured prototype pin (**B**) under 30° angle of vision captured with confocal laser scanning microscopy. The spatial surface topology is shown in (**C**) whereby the white dashed box in (**B**) shows the position of the scan.

**Figure 7 biomimetics-05-00014-f007:**
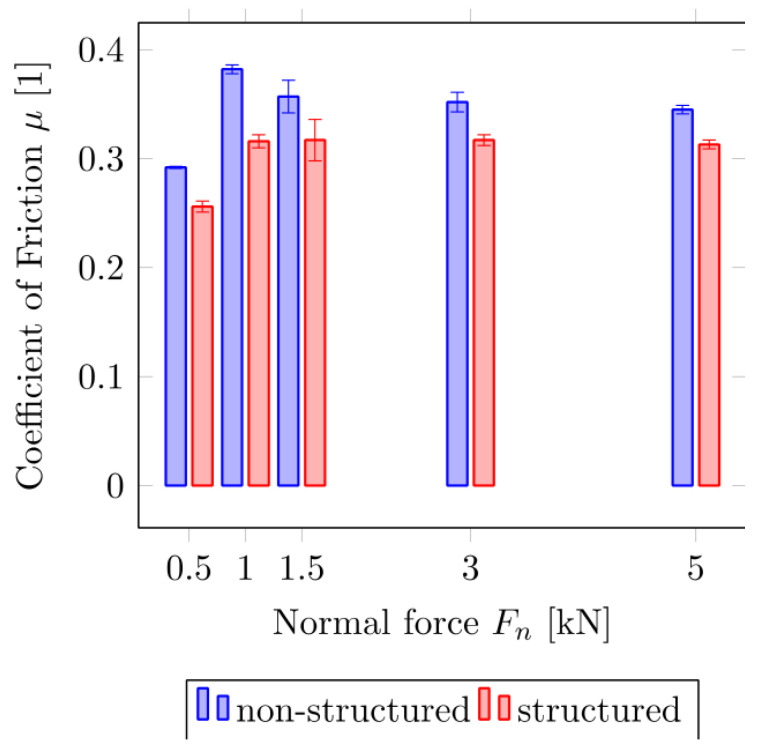
Results of the friction measurement on the tribometer for the non-structured (blue) and structured (red) samples. Error bars show standard deviations of the means measured for 3 individual pins.

**Figure 8 biomimetics-05-00014-f008:**
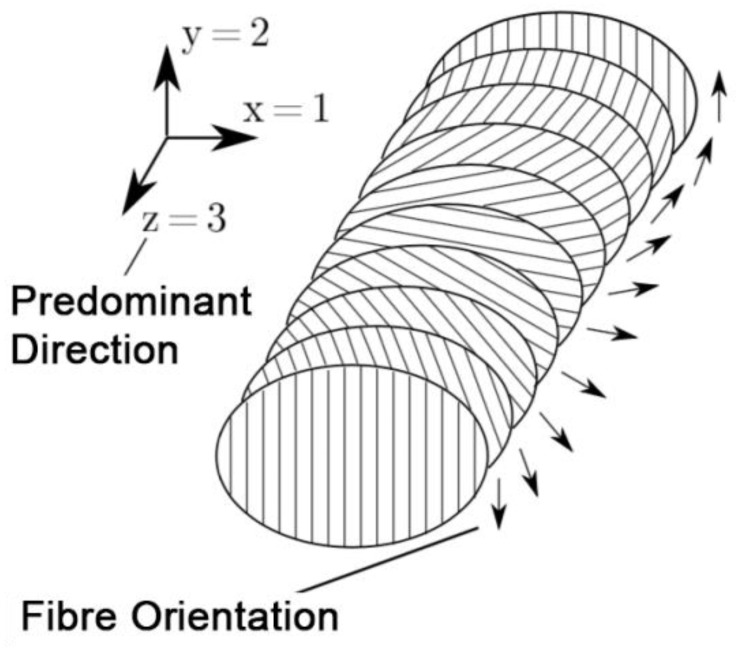
Transversal isotropic material architecture under consideration of Bouligand helix model of spatial orientation of the chitin fibres in the procuticle.

**Table 1 biomimetics-05-00014-t001:** Used material parameters for modeling the transversal isotropic material properties of *P. marginata*. E_1_, E_2_, E_3_: Young’s moduli; nu_12_, nu_13_, nu_23_: Poisson ratio; G_12_, G_13_, G_23_: moduli of shearing.

Parameter	Tibia	Tarsus	Reference
E1	3090 MPa	705 MPa	Equation 2
E2	3090 MPa	705 MPa	Equation 2
E3	2060 MPa	470 MPa	[18]
nu12	0.2	0.2	[22]
nu13	0.3	0.3	[23]
nu23	0.3	0.3	[23]
G12	1287.5 MPa	293.75 MPa	Equation 2
G13	792 MPa	181 MPa	Equation 2
G23	792 MPa	181 MPa	Equation 2

**Table 2 biomimetics-05-00014-t002:** Thickness of the cuticle layers of the tibial–tarsal joints of *P. marginata* (average values with standard deviations, *n* = 5).

	Epi (µm)	Exo (µm)	Endo (µm)	Exo (µm)
TI_d_	1.02 ± 0.17	12.14 ± 0.35	73.20 ± 3.40	7.72 ± 0.93
TA_p_	0.91 ± 0.13	11.81 to 22.49	56.37 ± 2.72	n.a.

Epi: epicuticle; Exo: exocuticle; Endo: endocuticle. TI_d_: the distal end and the condyle of the tibia; TA_p_: the proximal end and the condyle of the tarsus.

**Table 3 biomimetics-05-00014-t003:** Geometric dimensions of the prototype pin surface measured with confocal laser scanning microscopy (Figure 6).

	Length (µm)	Width (µm)	Height (µm)
Pad	127.65 ± 0.7	60.9 ± 0.6	96.17 ± 0.4
Groove	-	67.67 ± 0.8	96.17± 0.4

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
