# Peer review of "Structure and Frictional Properties of the Leg Joint of the Beetle Pachnoda marginata (Scarabaeidae, Cetoniinae) as an Inspiration for Technical Joints"

_biomimetics, 2020, doi:10.3390/biomimetics5020014_

Round 1

Reviewer 1 Report

I am content with the modifications the authors have made.

Author Response

Thank you very much for the review of our paper. I wish you and your family happy Easter.

Reviewer 2 Report

Vagts et al. report effects of asymmetric microstructures on friction force to understand leg kinematics of beetle. While the theme is interesting and main claim is supported to some extent by the experiment and FEA simulation, presentation is not carefully prepared and experimental data are not fully explained by the discussions.

- Authors showed changes in frictions before and after drying of specimen. Are there any dimensional changes in microstructures before and after drying? Provide quantified results by confocal scanning microscopy

- Authors mentioned “When the normal force is in the range of mN and μN, capillary, electrostatic, and Van der Waals forces play not a negligible role, but might dominate in their contribution to friction” in Introduction. In this study, measured force is in the order of sub mN where capillary, electrostatic, and Van der Waals forces play not a negligible role, As the fresh and dried samples had significantly lower coefficient of friction, there might be contribution from capillary force from the surface water on the contact or electrostatic attraction.   

- Figure 7. Does it have anisotropic surface roughness originating from machining direction? Provide, confocal scanning microscopy results for Fig 7A as well.

Provide quantified areas of microstructures (i.e. fill factor) so that future research can compare the result with this research.

- In Figure 9, material properties can be isotropic through the transverse direction if 180 deg (or the half the helix pitch) condition is met. However, as the helix is intrinsically chiral, I was wondering if there are direction dependent anisotropic of friction according to the helix chirality.  

- Regarding the alignment dependent mechanical properties, I recommend to cite below reference

ACS Appl. Mater. Interfaces 2016, 8, 12, 8110-8117

- In 4.2 authors mentioned possible trapping of wear particles in grooves. Present trapped dirt after the tribology tests with repeated cycles and present tribology results with repeated cycles. Then, show the tribology data and visualized images after washing the dirt.

It seems that this manuscript is not carefully prepared by authors based on various errors and typos.

  1. The manuscript has no Figure 1.
  2. Subscripted s is necessary for Fs in “The static frictional force Fs must be overcome, in order to set two contacting objects in motion”.
  3. Femurtibial: typo of femorotibial joint?
  4. “Tibial tarsus joint” should read as tibial tarsal joint
  5. “Due to the FEA-analysis”: FEA stands for finite element analysis. So, it is saying analysis twice.
  6. “Figs. 2A” should read as Fig. 2A.
  7. “The fiber bundles running perpendicular and parallel to the fracture plane are clearly visible in Figs. 2A.”: Do you really mean Fig. 2A?
  8. Figure 2B and C: Scale bars are missing.
  9. Figure 7. A: Scale bar is missing.
  10. Include information on confocal scanning microscopy in Materials and Methods section.

Author Response

Vagts et al. report effects of asymmetric microstructures on friction force to understand leg kinematics of beetle. While the theme is interesting and main claim is supported to some extent by the experiment and FEA simulation, presentation is not carefully prepared and experimental data are not fully explained by the discussions.

  • Authors showed changes in frictions before and after drying of specimen. Are there any dimensional changes in microstructures before and after drying? Provide quantified results by confocal scanning microscopy
    • Text added to chapter 4.1, p.11: We did not study this directly. However, from other literature sources we know that the shrinkage of highly-sclerotized areas of cuticle is not strong. The shrinkage can cause some additional waviness of the surface (as previously reported by Barbakadze et al. (2006) and Barbakadze (2019)) due to the stronger deformations of inner (stronger-hydrated layers). However, the superficial layers, which are responsible for the formation of the microstructures, which were studied in the paper under consideration, almost do not shrink. These shrinkages are usually at the nanoscale as it was previously shown by studies on the optical properties of cuticle layers. See for example: Fossilized Biophotonic Nanostructures Reveal the Original Colors of 47-Million-Year-Old Moths. Maria E. McNamara, Derek E. G. Briggs, Patrick J. Orr, Sonja Wedmann, Heeso Noh, Hui Cao. PlosOne, 2011, doi.org/10.1371/journal.pbio.1001200.
  • Authors mentioned “When the normal force is in the range of mN and μN, capillary, electrostatic, and Van der Waals forces play not a negligible role, but might dominate in their contribution to friction” in Introduction. In this study, measured force is in the order of sub mN where capillary, electrostatic, and Van der Waals forces play not a negligible role, As the fresh and dried samples had significantly lower coefficient of friction, there might be contribution from capillary force from the surface water on the contact or electrostatic attraction.
    • The text in chapter 4.1, p.11, was changed and the contribution of capillary and electrostatic forces was mentioned.
  • Figure 7. Does it have anisotropic surface roughness originating from machining direction? Provide, confocal scanning microscopy results for Fig 7A as well.

    • Contact surface of all samples were randomly grounded with a target roughness of Rz < 3 μm and quality controlled (added to chapter 2.7). Then the structured samples were laser micromachined as described in chapter 2.7 before friction measurement applied. Unfortunately, the original scans made by confocal scanning microscopy are not available anymore (just the data on roughness).
  • Provide quantified areas of microstructures (i.e. fill factor) so that future research can compare the result with this research.
    • Percentage of contact surface is calculated to 31% in comparison to a non- structured surface. This is added to chapter 3.2.
  • In Figure 9, material properties can be isotropic through the transverse direction if 180 deg (or the half the helix pitch) condition is met. However, as the helix is intrinsically chiral, I was wondering if there are direction dependent anisotropic of friction according to the helix chirality.
    • The text in chapter 4.1, p.11, was changed and the possible effect of helix chirality to friction was mentioned.
  • Regarding the alignment dependent mechanical properties, I recommend to cite below reference: ACS Appl. Mater. Interfaces 2016, 8, 12, 8110-8117
    • The results of molecule alignment studies were added to chapter 4.1, p.11

  • In 4.2 authors mentioned possible trapping of wear particles in grooves. Present trapped dirt after the tribology tests with repeated cycles and present tribology results with repeated cycles. Then, show the tribology data and visualized images after washing the dirt.
    • Text was added in chapter 2.5, p.4: Before and after friction measurement samples were cleaned in ultrasonic bath for five minutes at 50 °C and degreased with Rivolta (A.C.S. Spray).

It seems that this manuscript is not carefully prepared by authors based on various errors and typos.

  1. The manuscript has no Figure 1.

    1. Done

  2. Subscripted s is necessary for Fs in “The static frictional force Fs must be overcome, in order to set two contacting objects in motion”.

    1. Done

  3. Femurtibial: typo of femorotibial joint?

    1. Done

  4. “Tibial tarsus joint” should read as tibial tarsal joint

    1. Done
  5. “Due to the FEA-analysis”: FEA stands for finite element analysis. So, it is saying analysis twice.

    1. Done

    6. “Figs. 2A” should read as Fig. 2A.

I changed it together with 7.

7. “The fiber bundles running perpendicular and parallel to the fracture plane are clearly visible in Figs. 2A.”: Do you really mean Fig. 2A?

I changed the reference to Fig. 1D-G

  1. Figure 2B and C: Scale bars are missing.

    1. Added

  2. Figure 7. A: Scale bar is missing.

    1. Added

10. Include information on confocal scanning microscopy in Materials and Methods section.

Chapter „2.3 Confocal Laser Scanning Microscopy“ was added to Materials and Methods chapter

Round 2

Reviewer 2 Report

Authors answered previously raised concerns and improved quality of the work by including new experimental data. Hence, I can recommend for publication of this manuscript in Biomimetics.

This manuscript is a resubmission of an earlier submission. The following is a list of the peer review reports and author responses from that submission.

Round 1

Reviewer 1 Report

This paper provides various aspects of the leg joint of beetle especially the tibia-tarsus joints. The results consist of nanotribometer measurements of biological samples, SEM images of the cross-section of the cuticle, and tribometer measurements of patterned metal pins.

The readers are forced to visit the papers from the same authors to understand the context that this paper wants to focus on the microstructure (protuberances) on the distal end of the tibia. Please improve the description of the hypotheses and provide an overview of the target contact pair. The paper should be comprehensible by itself.

In the conclusion, the authors claim that "we show that the coefficient of sliding friction has a dependence on the surface microstructure and that a directional dependence of the mu_k is given by the asymmetric microstructure." However, the reviewer cannot find the corresponding results to support the claims.

The FEM analysis of microstructure is just a series of colorful images and no effort to explain the results of nanotribometer measurements. There is no significant association between different directions. Plus, the anisotropic microstructure of the biological system cannot be transferred in the experiment with a pin-on-disc tribometer.

Please show the statistically significant difference between distal and proximal directions to verify the anisotropy of the coefficient.

If the author wants to claim that the deformation of the microstructure explains the measured friction properties, for example, FEM experiments with and without microstructure are required. The results shown in Fig.3 could be simply explained by the surface properties of the cuticle without considering microstructures.

The observations of cuticle structure and tribological experiments are discussed in parallel without logical connections. The authors imply that the material properties should be analyzed for a better understanding of the insect joints. However, the relationships between the coefficient of sliding friction and the transversal isotropic model of the microstructure are not verified. Please provide a rationale for embracing anisotropic models.

It seems that the machined texture in the tribometer experiment is different from biological protuberances in terms of dimensions.

Minor comments:

Please use a multiplication sign instead of a letter x.

Reviewer 2 Report

I am surprised that the author choose as their initial reference on tribology the book by Rabinowicz – this was published initially in 1965 – and things have moved on a lot since then. The authors quite erroneously state that ‘friction is usually caused by the mechanical interlocking of the asperities of both contacting surfaces’. This is wrong, mechanical interlocking is an extremely rare occurrence. The slopes or real asperities, even on the roughest of machine surfaces, are only a fraction of a degree – the sources of friction, the physical mechanisms that impede motion, at the macroscale are just the same as in small scale devices — conventionally designated the ‘adhesion’ and ‘deformation’ terms — what can be different are the number and statistical properties of the asperity distributions. Asymmetrical friction can of course arise or be generated by suitable surface structures, for example by surface features resembling fish scales or roof slates, but these are not asperities in the usual sense of the word.

The paper falls into three parts. A careful experimental study of the symmetric friction exhibited between the tibia and tarsus of the beetle Pachnoda marginata. Even allowing for the natural variation of values there is a clear difference, of around 12%, in the coefficients of friction measured in the two opposing directions of sliding motion, distal > proximal. The authors illustrate the complex and non-isotropic sub-surface structure and directional surface topography that are associated with this effect. They then generate a numerical model of a single ‘shark-fin’ shaped microstructural feature and simulate the normal approach of this feature onto a planar surface to illustrate the changes in the geometry. The opposing half-space would seem to be rigid, is that right? And the relevance of this normal approach to asymmetric friction, static or dynamic, is unclear and unexplained.

Finally, the authors prepare a textured surface, is this PEEK? And compare dry friction values against a steel counterface in a pin-on-disc set-up. I don’t see the relevance of this to rest of the paper particularly when the authors write ‘The anisotropic friction properties of the biological system are not transferred to the technical samples’. This section adds nothing.